# A multi-centennial record of past floods and earthquakes in Valle d'Aosta, Mediterranean Italian Alps

Bruno Wilhelm[1], Hendrik Vogel[2], Flavio S. Anselmetti[2]

[1]University Grenoble Alpes, CNRS, IRD, Institut des Géosciences et de l'Environnement, F-38000 Grenoble, France

[2]Institute of Geological Sciences and Oeschger Centre for Climate Change Research, University of Bern, CH-3012 Bern, Switzerland

*Correspondence to: B. Wilhelm (bruno.wilhelm@univ-grenoble-alpes.fr)*

**Abstract**

Mediterranean Alpine populations are particularly exposed to natural hazards like floods and earthquakes because of both the close Mediterranean humidity source and the seismically active Alpine region. Knowledge of long-term variability in flood and earthquake occurrences is of high value since it can be useful to improve risk assessment and mitigation. In this context, we explore the potential of a lake-sediment sequence from Lago Inferiore de Laures in Valle d'Aosta (Northern Italy) as a long-term record of past floods and earthquakes. The high-resolution sedimentological study revealed 76 event layers over the last ca. 270 years; 8 are interpreted as most probably induced by earthquakes and 68 by flood events. Comparison to historical seismic data suggests that the recorded earthquakes are strong (epicentral MSK intensity of VI-IX) and/or close to the lake (distance of 25-120 km). Compared to other lake-sediment sequences, Lago Inferiore de Laures sediments appear to be regionally the most sensitive to earthquake shaking, offering a great potential to reconstruct the past regional seismicity further back in time. Comparison to historical and palaeoflood records suggests that the flood signal reconstructed from Lago Inferiore de Laures sediments well represents the regional and (multi-)decadal variability of summer-autumn floods, in connection to Mediterranean mesoscale precipitation events. Overall, our results reveal the high potential of Lago Inferiore de Laures sediments to extend the regional earthquake and flood catalogues far back in time.

**Key-words:** sediment record, earthquake, flood, century, Mediterranean Alps

## 1. Introduction

Natural hazards (e.g. earthquakes, floods, landslides, etc.) are of particular concern for societies as they cause widespread loss of life, damage to infrastructure and economic deprivation (e.g. Münich Re Group, 2003). The frequency of both geological (i.e. earthquakes) and hydrological (i.e. floods) events vary in time mainly as a function of tectonic processes and climatic regimes, respectively. Such long-term changes need to be taken into account for more accurate risk assessments. This becomes even more crucial in the context of global warming,

which is expected to lead to a modification of the hydrological cycle and associated floods (IPCC, 2013). However, available instrumental time-series generally cover a short time span, precluding a comprehensive knowledge of the tectonic and climatic-related variability. In this respect, historical and natural archives have been widely studied to extend earthquake and flood catalogues further back in time (e.g. Guidaboni et al., 2007; Rizza et al., 2011; Brázdil et al., 2012; Ballesteros-Cánovas et al., 2015; Benito et al., 2015; Denniston et al., 2015; Ratzov et al., 2015). Among them, lake sediments have shown to be valuable archives as they record past events in a continuous and high-resolution mode (e.g. Lauterbach et al., 2012; Wilhelm et al., 2012a; Strasser et al., 2013; Wirth et al., 2013; Amman et al., 2015; Van Daele et al., 2015). The greater hydraulic energy of flooded rivers increases their capacity to erode and transport sediments. Downstream, lakes may act as sediment traps, resulting in the deposition of a coarser-grained layer that will be preserved in time (e.g. Gilli et al., 2013; Schillereff et al., 2014). In the case of earthquakes, ground shaking may disturb lake sediments by triggering co-seismic in situ deformation or post-seismic deposits related to subaquatic mass movements of slope sediments and resuspension (e.g. Avşar et al., 2014; Van Daele et al., 2015). Identification and dating of all 'event layers' in sediment cores enable to reconstruct past event occurrences over centuries to millennia. Recently, some studies have also developed methods to reconstruct the magnitude of past events. The magnitude of past flood events may be reconstructed through grain size (e.g. Schiefer et al., 2011; Lapointe et al., 2012; Wilhelm et al., 2015; Schillereff et al., 2015) or through the total volume of sediments transported and deposited during the flood (e.g. Jenny et al., 2014; Wilhelm et al., 2015). Reconstruction of past earthquake magnitudes and location is approached by comparing regional records of seismic-induced deposits (e.g. Strasser et al., 2006; Wilhelm et al., 2016b) or through the deposit's spatial extent and thickness (Howarth et al., 2014; Moernaut et al., 2014).

The southern European Alps (Northern Italy) are particularly harmed by natural hazards such as floods and earthquakes (e.g. Boschi et al., 2000; Guzetti and Tonelli, 2004; Eva et al., 2010), due to the proximity of both the Mediterranean Sea and the seismically-active Alpine region. The Mediterranean Sea is the primary moisture source for orographic precipitation on the southern flank of the Alps (e.g. Buzzi and Foschini, 2000; Lionello et al., 2012). Spatially restricted convective and spatially more exhaustive cyclonic precipitation events may lead to catastrophic floods (Gaume et al., 2009; Marchi et al., 2010), as for instance occurred in October 2000 or June 1957 (Ratto et al., 2003). Moreover, the south-western European Alps is a seismogenic region that experienced strong earthquakes with macroseismic Medvedev-Sponheuer-Kárník (MSK) intensities up to IX and estimated magnitudes higher than 6., e.g. the Ligurian earthquake in 1887 (Mw = 6.8; Larroque et al., 2012) and the Visp earthquake in 1855 (Mw = 6.2; Fäh et al., 2011; Fig. 1),

In this context, the present study aims at exploring the potential of a lake sequence as recorder of past floods and earthquakes in the western Italian Alps. This is undertaken by studying the high-elevation sediment sequence of Lago Inferiore de Laures, Valle d'Aosta.

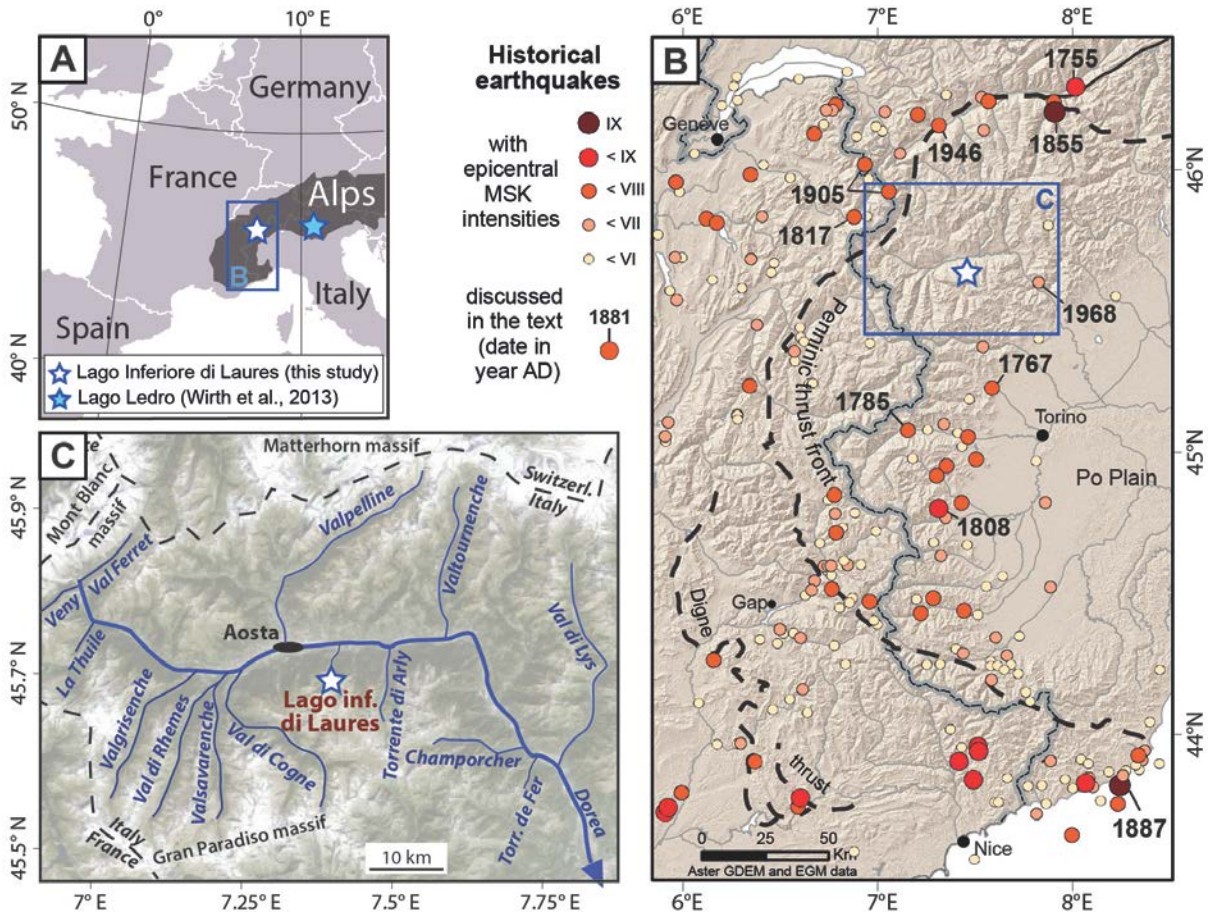

**Figure 1.** (A) Location of Lago Inferiore de Laures in the Mediterranean Italian Alps, with (B) locations of historical earthquakes with epicentral MSK intensity above IV. The earthquake catalog is provided by the database SisFrance (http://infoterre.brgm.fr/; Lambert and Levret-Albaret, 1996; Scotti et al., 2004). (C) Location of Lago Inferiore de Laures catchment area in the hydrological network of Vallee d'Aosta that is regularly affected by floods as documented by Mercalli et al. (2003).

## 2. Study site

Valle d'Aosta is located at the foot of the Mont Blanc and Monte Rosa massifs, north to the vast Italian Po Plain, and ~180 km north of the Mediterranean Sea (Fig. 1). Lago Inferiore de Laures (2450 m a.s.l., 45°41'N, 7°24'E) is a small, high-elevation lake located on the north-facing slope of Vallee d'Aosta (Fig. 1C). Due to the high elevation of the catchment, only the area surrounding the lake is covered by alpine meadow vegetation, which is impacted by grazing activity. Most of the catchment is covered by bedrock and scree. Rock is mainly made of eclogitic micaschist, which was eroded by small glaciers in the western and southern parts of the catchment as evidenced by the presence of glacial deposits and moraines (Fig. 2). These glaciers have disappeared and only a rock glacier is still active in the south-eastern part of the catchment. The catchment is mainly drained by the mountain stream that crosses Lago Superiore and Lago Lungo before entering Lago Inferiore. These two upper lakes act as two sediment traps and, thereby, all the upper part of the catchment barely contributes to the detrital inputs in Lago Inferiore. Detrital inputs are mainly provided by (i) the lower part of the main stream and its eastern tributary and (ii) a temporary stream that drains glacial deposits west from the lake. This results in two

distinct major detrital input sources to the lake, as suggested by the aerial and subaquatic deltas built on the
eastern and western lake shores. Mobilization of detrital material is restricted to summer months and beginning
of autumn (June/July to mid-November) when the lake ice cover is absent and catchment soils are thawed and
free of snow cover.

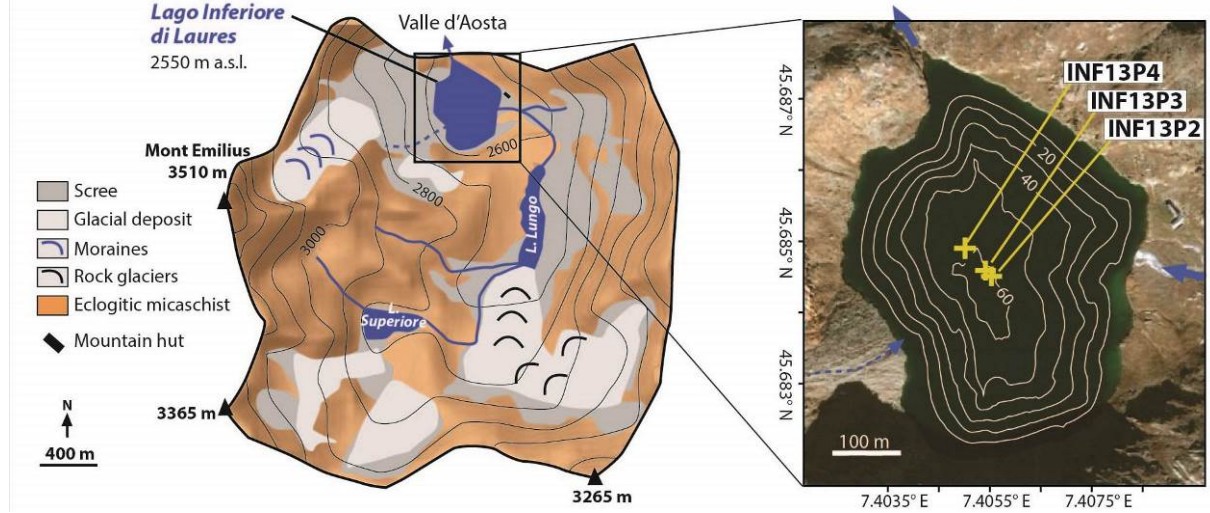

**Figure 2.** Geological and geomorphological characteristics of the Lago Inferiore de Laures catchment area (left panel).
Bathymetric map of Lago Inferiore de Laures and coring sites (right panel).

**3. Methods**
**3.1. Core description and logging**
In fall 2013, a bathymetric survey with a single-beam echosounder was carried out at Lago Inferiore and
revealed a narrow flat basin in the centre of the lake with a maximum water depth of 60.7 m (Fig. 2). Three up to
62 cm long gravity cores have been retrieved from the depocenter of the lake. The uppermost 13 cm of core
INF13P2 were disturbed during the coring. The three cores were split lengthwise and the visual macroscopic
features of each core were examined in detail to determine the different sedimentary facies. Based on these
facies, a stratigraphic correlation was carried out between the three cores to document the spatial extent and
succession of the different facies over the lake basin.
High-resolution images and gamma-ray attenuation bulk density (GRAPE) data were acquired on a Geotek$^{TM}$
multisensor core-logger (Institute of Geological Sciences, University of Bern). The bulk density is obtained at a
5-mm downcore resolution. X-Ray analyses on the core INF13P3 were carried out on an Itrax™ (Cox Analytical
Systems) X-ray fluorescence (XRF) core scanner (Institute of Geological Sciences, University of Bern), using a
Molybdenum tube, set to 30 kV, 35 mA with a 10-s count-time and a 1-mm sampling step. The scattered
incoherent (Compton) radiation of the X-ray tube ($Mo_{inc}$) varies with bulk element mass/sediment density
(Croudace et al., 2006) and, thereby, provides a high-resolution proxy for sediment density (e.g. Wilhelm et al.,
2016a). $Mo_{inc}$ values were averaged at a 5-mm resolution for correlation with the GRAPE-density, which
resulted in a linear, positive, and significant correlation (r=0.88, $p<10^{-4}$). This allowed using $Mo_{inc}$ as a proxy of
sediment density for identifying mm-scale event layers, e.g. flood and mass-movement deposits. Event layers are
characterized by higher density because of the high amount of detrital material provided in a short time (e.g.
Støren et al., 2010; Gilli et al., 2012; Wilhelm et al., 2012b).
Grain-size analyses were performed on core INF13P3 using a Malvern Mastersizer 2000 (Institute of Geography,
University of Bern) at a 5-mm continuous interval. Before the grain-size analysis, the samples have been treated
in a temperate bath of diluted (30%) hydrogen peroxyde during 3 days to remove the organic matter. The
disappearance of the organic matter was checked through smear slide observations. Grain-size analyses of the
detrital material were performed to characterize the transport-deposition dynamics of the deposits (e.g. Passega,
1964; Wilhelm et al., 2013; 2015).

**3.2. Dating methods**
To date the lake sequence over the last century, short-lived radionuclides ($^{226}$Ra, $^{210}$Pb, $^{137}$Cs) were measured by
gamma spectrometry at EAWAG (Dübendorf, Switzerland). The core INF13P3 was sampled following a non-
regular step of $1 \pm 0.2$ cm, matching the facies boundaries. The $^{137}$Cs measurements generally allow two main
chronostratigraphic markers to be located: the fallout of $^{137}$Cs from atmospheric nuclear weapon tests
culminating in AD 1963 and the fallout of 137Cs from the Chernobyl accident in AD 1986 (Appleby, 1991).
$^{226}$Ra is measured as a proxy for the supported $^{210}$Pb in order to calculate the unsupported $^{210}$Pb that corresponds
to the excess $^{210}$Pb (e.g. Schmidt et al., 2014). The decrease in excess $^{210}$Pb ($^{210}$Pbex) and the Constant
Flux/Constant Sedimentation (CFCS) model allow a mean sedimentation rate to be calculated (Goldberg, 1963).
The standard error of the linear regression of the CFCS model is used to assess the uncertainty of the
sedimentation rate. The $^{137}$Cs chronostratigraphic markers are then used to control the validity of the $^{210}$Pb-based
sedimentation rate.
In addition to short-lived radionuclides, historical lead (Pb) contaminations were also used to control the $^{210}$Pb-
based chronology (e.g. Renberg et al. 2001). In order to identify lead contamination, we used the geochemical
measurements carried out on the Itrax™ XRF core scanner on core INF13P3. Pb intensities were normalized to a
well-measured detrital element, i.e. titanium (Ti), to disentangle natural and human-induced changes in Pb.
Recorded Pb variations were compared to historical lead emissions in Switzerland (Weiss et al., 1999), the
closest place to the studied site where lead emissions are well documented.

**4. Results**

**4.1. Description of the sedimentary deposits**
The sediment consists of a finely bedded, greenish brown mud mainly composed of detrital material with grain
sizes in the silt-clay fraction and amorphous organic matter. Smear-slide observations reveal that the organic
matter content increases upcore, concurrently to the dark brown colour of these deposits (Fig. 3). These fine-
grained deposits, representing the background hemi-pelagic sedimentation, are interrupted by 77 beds
characterized by rather coarse material, lower organic matter content, and higher density. According to several
studies providing a comprehensive overview of event layers (e.g., Mulder and Cochonat, 1996; Gani, 2004; Van
Daele et al., 2015; Wilhelm et al., 2016), the 77 beds represent short-term depositional events and they
correspond to 74 graded beds (GBs), 1 matrix-supported bed (MSB), 1 homogeneous bed (HB) and 1 deformed
layer (Fig. 3).
The 74 GBs are all characterized by a sharp and coarse-grained base, a fining-upward trend and a thin, whitish
fine-grained capping layer. There is no evidence for erosive bases. The stratigraphic correlation reveals that
almost all GBs appear in the three cores. Only four GBs identified in cores INF13P3 (33.3-35 cm) and INF13P4
(29.6-32 cm) are missing in core INF13P2. In core INF13P2, the four missing GBs stratigraphically correspond
to a deformed layer (28-30 cm; Fig. 3).

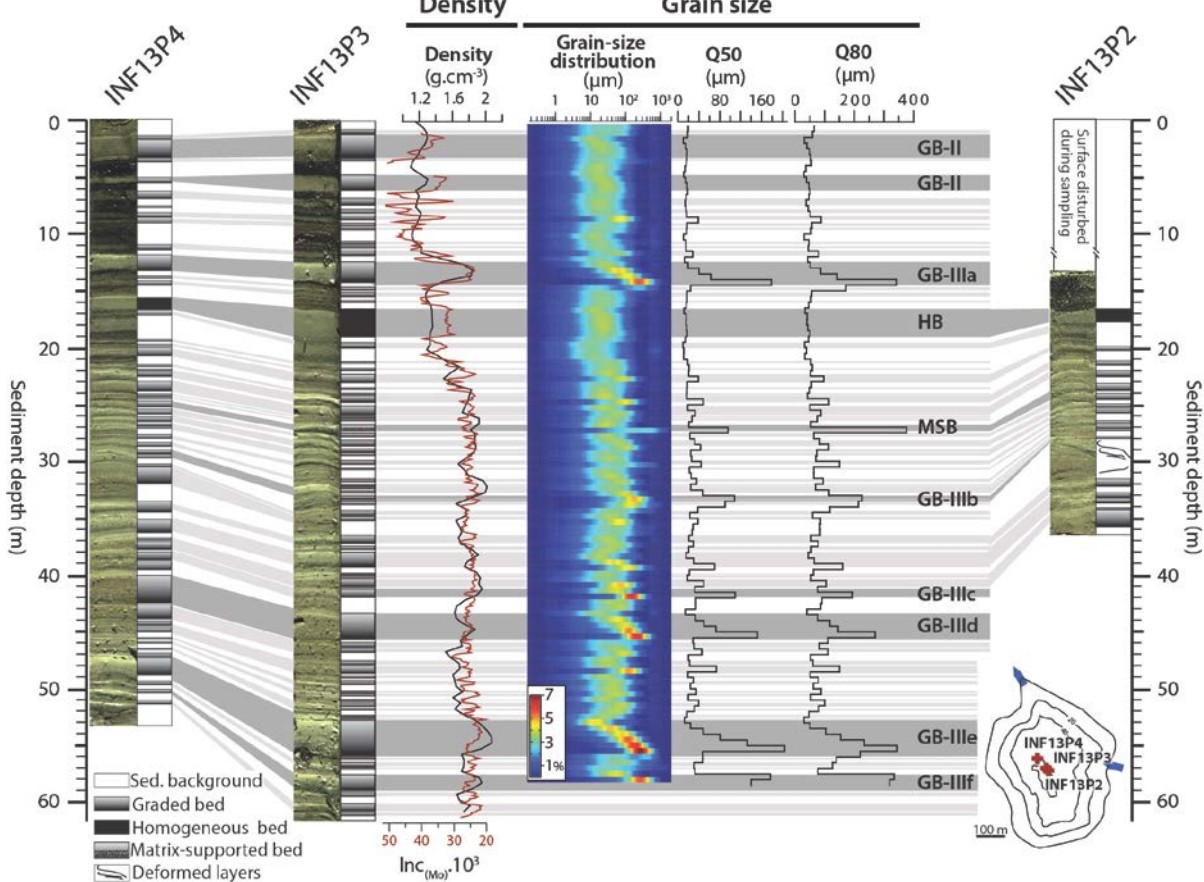


**Figure 3.** Lithological descriptions of cores and stratigraphic correlations based on sedimentary facies. Variability in grain-
size distribution is shown for the core INF13P3. The density measurements performed by gamma-ray attenuation are shown
close to Mo$_{inc}$, used as a high-resolution density proxy. The horizontal bars highlights the stratigraphical correlation between
cores with a distinction between two probable triggers of deposits (light versus dark grey) as discussed in sections 5.1. and
168 5.2.


The Passega-type (D50 *vs.* D80) diagram highlights a steady decrease of both the median (D50) and the coarse
percentile (D80) from the base to the top of the GBs (Fig. 4A). This confirms the visually-identified fining-
upward trend of all GBs. 'D50$_{max}$ *vs.* deposit thickness' and 'D80$_{max}$ *vs.* deposit thickness' diagrams (where
D50$_{max}$ and D80$_{max}$ are defined as the highest value of D50 and D80 of each GB) suggest that the 74 GBs may be
differentiated in 3 types (Fig. 4A). Most of the GBs (66 of 74) form a well-grouped cluster characterized by low
values of thickness (1 - 10 mm), $D50_{max}$ (10 - 50 µm) and $D80_{max}$ (35 - 115 µm). These 66 GBs are labelled GB-I
(dark blue points, Fig 4A). These diagrams highlight 2 GBs, labelled GB-II (light blue points, Fig. 4A), also
characterized by a very fine grain size ($D50_{max}$ of 16-18 µm and $D80_{max}$ of 50-52 µm) but a larger thickness (14-
21 mm) than GB-I. As a result, GB-II is characterized by an intermediate pattern between GB-I and HB. In
contrast, some GBs (6 of 74; labelled GB-III; red points, Fig. 4A) are scattered in the 'percentile *vs.* thickness'
diagrams but well distinguishable from GB-I and GB-II because of both their coarser grain size (D50 of 70 - 200
µm and D80 of 160 - 350 µm) and larger thickness (from 3 to 33 mm). The distinct characteristics of the three
GB types suggest distinct dynamics of sediment transport and deposition and, thereby, distinct triggers
(discussed in sections 5.1.1. and 5.2.1.).

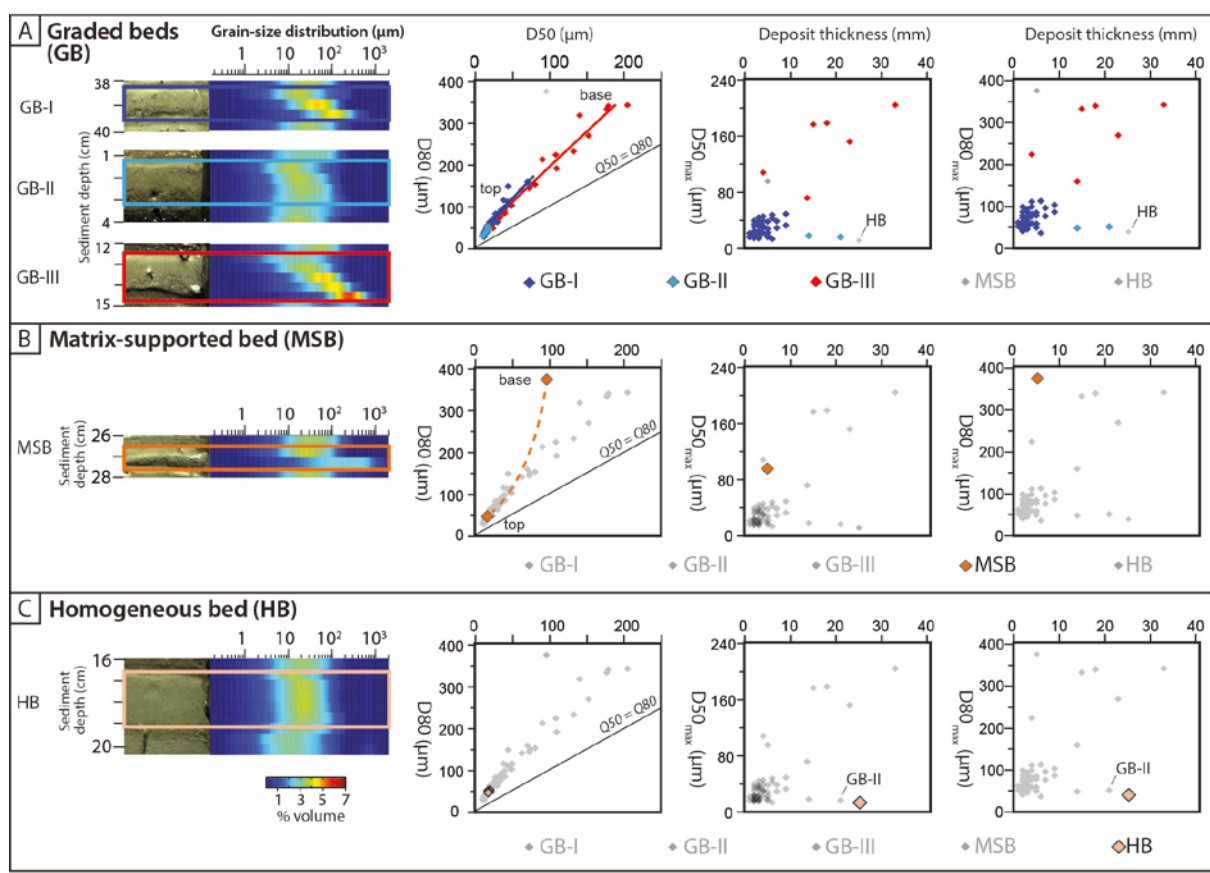


**Figure 4.** Close-eye views of event layers (left) and their positions in a Passega-type (Q50 vs. Q80) diagram as well as in
'percentile vs. deposit thickness' diagrams (right) for the graded beds (A), the matrix-supported bed (B) and the homogenous
bed (C).

The MSB identified at 27 cm in core INF13P3 differs from the GBs by the poorly sorted fining-upward trend
(Fig. 3 and 4B). This is well highlighted in the Passega-type diagram where the pattern is almost vertical,
describing a large decrease of the coarse percentile (D80) with much less variation of the median (D50). The 2.5
mm-thick HB identified at 17 cm in core INF13P3 is characterized by a sharp base, a thin, whitish fine-grained
capping layer and a central part with a fine and perfectly homogeneous grain size (Fig. 3 and 4C).
A 3.5 cm-thick layer at 28 cm in core INF13P2 is characterized by mixed beds in the lower part and folded beds
in the upper part (Fig. 3). The stratigraphic correlation reveals that this deformed layer is overlain by a thin
graded bed that becomes thicker in cores INF13P3 and INF13P4. In core INF13P3, this graded bed corresponds
to a GB-III (labelled GB-IIIb in Fig. 3). In addition, the stratigraphic correlation suggests that this deformed
layer is not intercalated in the sediment sequence (e.g. slump) but corresponds to in situ deformation.

**4.2. Chronology**
The excess $^{210}$Pb ($^{210}$Pbex) profile in cores INF13P3 shows a steady decrease downcore in activity from 436 to
11 Bq/kg. The profile is, however, punctuated by depths with very low values, which correspond to thick event
layers (Fig. 5). We excluded $^{210}$Pbex values associated with these instantaneous deposits to construct a synthetic
sediment record (Arnaud et al., 2002). The CFCS model (Goldberg, 1963) was applied to the synthetic $^{210}$Pbex
profile and indicates that the sequence is characterized by two periods of different sedimentation rates (SR). SR
shifts from $1.1 \pm 0.2$ mm.yr$^{-1}$ in the lower portion of the core to $1.4 \pm 0.36$ mm.yr$^{-1}$ in the topmost 5.5 cm. The
CFCS model-derived ages were used to develop continuous age-depth relationships for core INF13P3 (Fig. 5). A
synthetic $^{137}$Cs profile was built and displays a progressive increase until a peak of 1400 Bq.kg$^{-1}$ at 9.5 cm (Fig.
5). Such high $^{137}$Cs values are unequivocal of the fallout associated to the 1986 Chernobyl accident in the region
(e.g. Vannière et al., 2013; Wilhelm et al., 2012a; Etienne et al., 2012; Wilhelm et al., 2016a). The second
expected peak related to the nuclear weapon tests in AD 1963 cannot be as clearly defined.

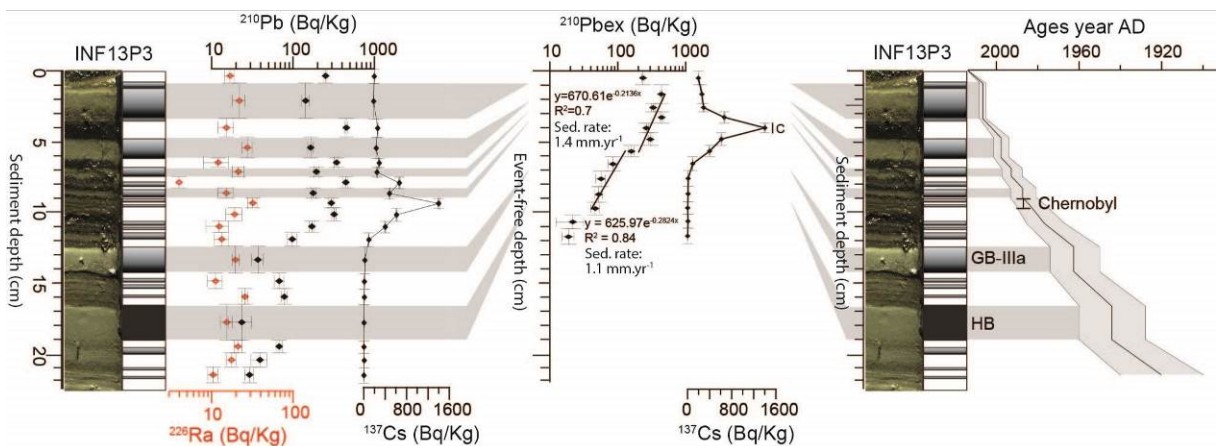


**Figure 5.** $^{226}$Ra, $^{210}$Pb and $^{137}$Cs profiles for core INF13P3 (left). Application of a CFCS model to the event-free sedimentary
profile of $^{210}$Pbex. Resulting age–depth relationship with 1σ uncertainties and locations of the historic $^{137}$Cs peak of
Chernobyl (AD 1986) supporting the $^{210}$Pb-based ages.

The Pb/Ti ratio shows a low background ($\leq 0.5$) in the lower part of core INF13P3 (Fig. 6). At 21 cm, the Pb/Ti
ratio increases and remains almost always above 0.5 upcore. From 13 to 8 cm, it reaches high values ($> 1$) with a
maximum at 10 cm ($> 4$). These distinct steps well mirror historical Pb emissions in Switzerland with low
emissions ($< 500$ tons.year$^{-1}$) until AD 1920 and high emissions ($> 1000$ tons.year$^{-1}$) from the 1950s to the
1980s, with a maximum around 1970 (Weiss et al., 1999). The increase of Pb emission in the 1920s may
correspond to the beginning of the use of leaded gasoline and the peak in Pb emission (1970s) to its maximal use
(Weiss et al., 1999; Arnaud et al., 2004). These two steps in historical Pb contaminations, well-marked in the
Pb/Ti ratio, may thus be used as additional chronological markers.
Overall, the good chronological agreement between these independent markers ([137]Cs peak and Pb peaks) and
the [210]Pb-derived ages supports the validity of our age-depth model (Fig. 7). The extrapolation of the CFCS
model-derived ages suggest that core INF13P3 covers the ~270 years (Fig. 7).

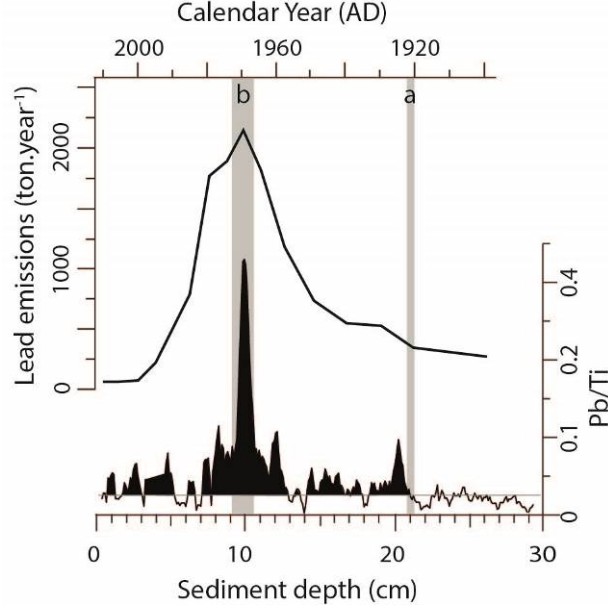

**Figure 6.** Historical lead (Pb) emissions in Switzerland (from Weiss et al., 1999) compared to the Pb/Ti ratio measured in core INF13P3.

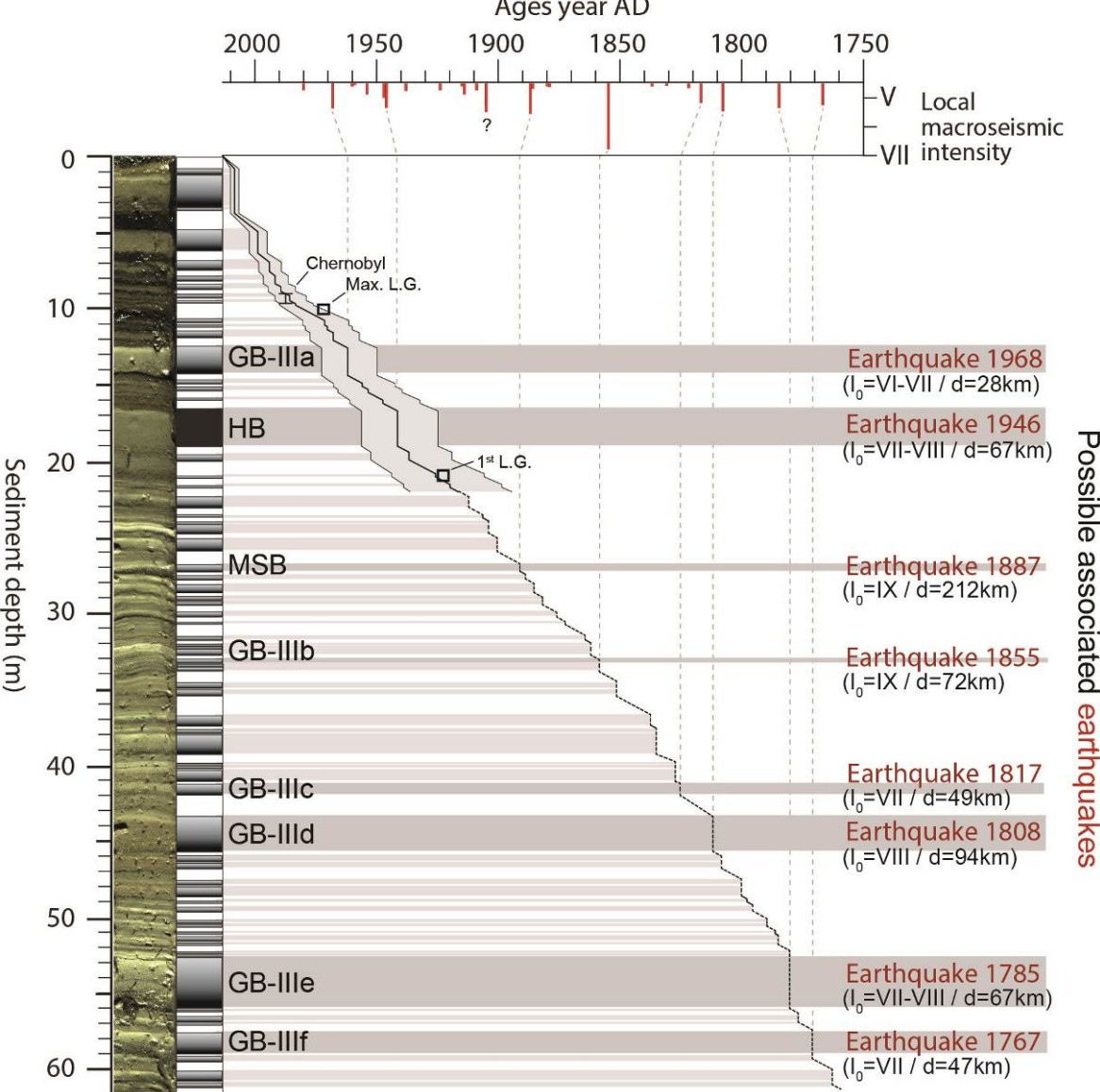

235

**Figure 7.** Age–depth relationship of core INF13P3 based on the [210]Pb-based sedimentation rate (with 1σ uncertainties) for the last century and based on the extrapolation of this sedimentation rate for the lower part of the core. The three chronological markers supporting the [210]Pb-based sedimentation are shown: the [137]Cs peak associated to the Chernobyl accident (AD 1986), the first use of leaded gasoline (1920s) and its maximal use (1970s). Labels (GB-III, HB and MSB) correspond to the mass-movement-induced deposits. Historical earthquakes, possibly associated to these mass-movement-induced deposits, are indicated with their respective epicentral MSK intensity ($I_0$) and their distance to the lake (d). The upper panel represent the seismic intensity triggered by the strongest and/or closest historical earthquakes in the lake area.

## 5. Discussion

### 5.1. Lago Inferiore de Laures sediments: a record of past earthquakes?

#### 5.1.1. Trigger of MSB, HB, GB-III and the deformed layer

The MSB pattern in the Passega-type diagram suggests that the transport energy is supplied by the sediment weight rather than by a water current velocity, i.e. formation of concentrated density flows of suspended

sediments during a sub-aquatic mass movement (e.g. Mulder et Cochonnat, 1996; Arnaud et al., 2002; Wilhelm
et al., 2016b). The HB characteristics are very similar to deposits previously described by many studies (e.g.
Schnellmann et al. 2005; Beck 2009, Petersen et al., 2014). These studies proposed that a sub-aquatic mass
movement triggers the oscillation of the whole lake water body (i.e. seiche), which homogenizes the sediment
put in suspension by either the water oscillation or the mass movement. Therefore, HB most probably results
also from a mass movement.
GBs are associated with turbidity currents triggered by either flood events or mass movements (e.g. Sturm and
Matter, 1978; Shiki et al., 2000; Arnaud et al., 2002; Mulder and Chapron, 2011; Wilhelm et al., 2012b). In the
latter case, they are formed by the sediment that is transported in suspension during the mass movement and then
deposited over the mass-wasting deposits and/or further in the lake basin (e.g. Shiki et al., 2000; Schnellmann et
al., 2005). These mass-movement-induced GBs are also known to be generally thicker than those induced by
flood events because mass movements may mobilize much larger quantities of sediments than floods (e.g. Shiki
et al., 2000; Schnellmann et al., 2005; Fanetti et al., 2008; Wilhelm et al., 2013). Accordingly, the rare GB-III
characterized by large thicknesses may be associated to mass movements. The position of GB-IIIb on top of the
deformed layer (Fig. 3) further supports this assumption because (i) the immediate stratigraphic succession
suggests a common trigger for these two deposits and because of (ii) the ability of strong earthquake shaking to
trigger (co-seismic) in situ deformation and (post-seismic) mass movements. Folded and mixed beds of the
deformed layer are similar characteristics to the so-called "mixed layers" that result from shear stress applied to
poorly consolidated sediments during strong earthquake shaking (e.g. Marco et al., 1996; Rodriguez-Pascua et
al., 2000; Migowski et al., 2004; Monecke et al., 2004). Accordingly, the deformed layer is interpreted as the
result of strong earthquake shaking. Because of the immediate stratigraphic succession with the GB-IIIb, these
two beds are interpreted as one event layer triggered by a common earthquake.

### 5.1.2. Chronological control on the mass-movement-induced layers

Mass movements can be triggered by spontaneous failures due to overloading of slope sediments, snow
avalanches, fluctuations in lake levels, rockfalls, or earthquakes (e.g., Van Daele et al., 2015; Wilhelm et al.,
2016b). Here changes of lake level can be excluded because the water level of Lago Inferiore is well controlled
by a bedrock outlet. Rockfalls seem also unlikely as there is no geomorphological evidence of major rockfalls in
the catchment. Earthquakes are known to affect the region and may thus be a good candidate. In addition, the
earthquake trigger is the only candidate to explain the in situ deformed layer with associated GB-IIIc. To test the
earthquake trigger of all mass-movement-induced layers (i.e. GB-III, HB and MSB), their ages are compared to
the dates of historical earthquakes well documented over the last centuries (database SisFrance,
http://www.sisfrance.net; Lambert and Levret-Albaret, 1996; Scotti et al., 2004 and database CFTI4Med,
http://storing.ingv.it/cfti4med/, Guidoboni et al., 2007). In addition to the chronological agreement, the
potentially recorded earthquakes are also expected to be the strongest and/or the closest to the lake, as those are
expected to have generated the largest ground motions in the lake area. To take into account this second
parameter, the seismic intensity of each historical earthquake in the lake area was estimated in first order by
using the following equation from Wilhelm et al. (2016b):
$$y = \alpha . \ln(x) + b,$$

where *x* corresponds to the distance between the lake and the epicenter, *y* to the epicentral intensity of the historical earthquakes, *α* to the slope of the attenuation curve fixed to 1,13 for the region, and *b* to the local seismic intensity. From this estimation, 9 earthquakes during the last 250 years triggered local seismic intensities above V (Fig. 7), i.e. intensities that may be strong enough to trigger seismically-induced deposits (e.g. Moernaut et al., 2014; Howarth et al., 2014; Van Daele et al., 2015; Wilhelm et al., 2016b). GB-IIIa and HB are dated to AD 1962 ±12 and 1941 ±16 years, respectively (Fig. 5). These dates correspond well to the two most recent and 'strongest' historical earthquakes occurring in AD 1968 and 1946 (Figs. 1 and 7). The extrapolation of the $^{210}$Pb-based sedimentation rate allows estimating ages of the older mass-movement-induced layers to AD 1891, 1859, 1826, 1811, 1780 and 1771 (Fig. 7). All of them correspond well to earthquakes expected to have triggered the largest ground motions in the lake area in AD 1887, 1855, 1817, 1808, 1785 and 1767. Age differences between deposits and associated historical earthquakes are lower than 5 years, except between GB-IIIc dated to AD 1826 and the AD 1817 Chamonix earthquake. Surprisingly, although as strong as the other earthquakes, the Chamonix earthquake (AD 1905) does not seem to have triggered a deposit. Overall, this good temporal agreement highly supports that mass-movement-induced layers may have been triggered by historical earthquakes.

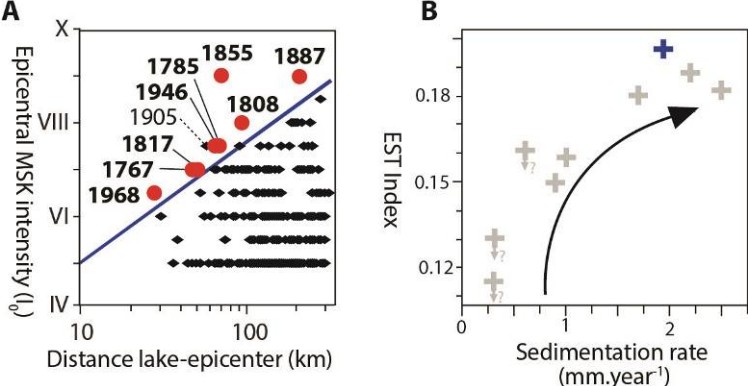

**Figure 8.** (A) Diagram "distance of earthquakes to the lake vs. epicentral MSK intensity" that aims at confirming that the earthquakes associated to the mass-movement-induced deposits are the strongest and/or the closest to the lake. Black crosses indicate all historic earthquakes closer than 120 km to the lakes with epicentral MSK intensities ≥ IV. Red dots with dates correspond to historical earthquakes associated to the mass-movement-induced deposits in Figure 7. The sensitivity threshold (blue line) is placed to delimit the recorded from the non-recorded earthquakes. (B) The 'Earthquake Sensitivity Threshold Index' (ESTI) is compared to the sedimentation rate for Lago Inferiore de Laures (blue cross) and other similar Alpine lakes (grey crosses) studied by Wilhelm et al. (2016b). Arrows show that these ESTIs are maximum values.

### 5.1.3. Earthquake record and lake sensitivity

The record of eight earthquakes over ~270 years (i.e. return period of ~35 years) suggests a high sensitivity of Lago Inferiore de Laures to earthquake shaking, as such a high frequency of earthquake-induced deposits has rarely been observed in the region (e.g. Guyard et al., 2007; Lauterbach et al., 2012; Simonneau et al., 2013; Strasser et al., 2013; Kremer et al., 2015; Chapron et al., 2016; Wilhelm et al., 2016b). All historical earthquakes are plotted in a 'distance vs. epicentral MSK intensity' diagram (e.g. Monecke et al., 2004; Wilhelm et al., 2016b) where the recorded earthquakes are highlighted in red (Fig. 8A). To quantify and compare its sensitivity

to other lakes, an empirical threshold line was defined that limits the domains of the recorded from the non-recorded earthquakes (blue line in Fig. 8A). The 'Earthquake Sensitivity Threshold Index' (ESTI), defined as the inverse of the intercept of this threshold line with the intensity axis at 10 km from the lake (Wilhelm et al., 2016b), offers a direct comparison of sensitivity with these other lakes is possible. The ESTI score for Lago Inferiore reaches 0.19, i.e. the highest value of the Alpine lakes for which the sensitivity was quantified (Fig. 8B). This high sensitivity of Lago Inferiore to earthquake shaking may be explained by many factors like slope angle, sediment thickness or geotechnical properties of the sedimentary succession (e.g., Morgenstern, 1967; Strasser et al., 2011; Ai et al., 2014; Wiemer et al., 2015). However, Wilhelm et al. (2016b) suggested that the dominant factor explaining the lake sensitivity of such Alpine lakes is the sedimentation rate, i.e. that the lake sensitivity increases when the sedimentation rate increases, which is in agreement with the lake's high sedimentation rate (Fig. 8B).

### 5.2. Lago Inferiore de Laures sediments: a record of past floods?

#### 5.2.1. Trigger of GB-I and GB-II

The high frequency of GB-I (66 deposits over 270 years, return period of ~4 years) makes it unlikely that these layers were the result of mass movements. In addition, the very uniform values of grain size and thickness characterizing GB-I suggest that they are triggered by processes where sediment erosion, transport and deposition are well controlled/regulated. Many studies suggested that the amount and grain size of eroded, transported and deposited material in case of flood events are controlled by the river discharge (e.g. Schiefer et al., 2011; Lapointe et al., 2012; Jenny et al., 2014; Wilhelm et al., 2015). Therefore, flood processes seem to be the best candidate to trigger GB-I.

The presence of grading in GB-II and their isolated positions in the 'percentile vs. thickness' diagram are similar characteristics to GB-III and HB, suggesting a common trigger for both GB-II and GB-III, i.e. mass movements. The two GB-II are dated to AD 2006 ±2 and 1997 ±4 yrs., respectively (Fig. 5). An earthquake trigger is very unlikely as no strong and/or close earthquake occurred at that time. A mass-movement trigger can, however, not be excluded. Alternatively, Giguet-Covex et al. (2011) suggested that thickness of flood-induced GBs may significantly increase without changes in grain size when human impact, i.e. grazing pressure in such high-elevation catchments, became high. Sheep grazing and trampling would accelerate the mechanical soil degradation, making erosion processes higher during floods. In this way, GB-II may also be triggered by floods at time of high grazing activity, which currently occurs close to the lake as evidenced by the sheep pen located on the shoreline of Lago Inferiore. In addition, these deposits appear in the uppermost part of the cores characterized by high organic matter content. This higher content of lacustrine organic matter might result from a higher primary production linked to an increase of nutrients inputs with the higher grazing activity.

#### 5.2.2. Chronological control on flood-induced deposits

The assignment of a flood trigger to GB-I and GB-II may be assessed by using historical flood data. A direct comparison between deposit ages and historical flood dates is precluded because the outlet stream of Lago Inferiore does not flow through any village downstream. Instead, the frequency of GB-I and GB-II occurrences

was compared to the frequency of historical summer-autumn floods that affected streams and villages around Lago Inferiore as documented by Mercalli et al. (2003). For the comparison, a historical flood event that occurred in mid-May (AD 1926) was not considered as we assume that the lake was frozen at that time. Over the last century, historical data reveal a high flood frequency (up to 4 floods per 11 years) during periods AD 1900-1920 and AD 1950-2010 and a low frequency (less than 1 flood per 11 years) during the period AD 1920-1950 (Fig. 9). This multi-decadal variability in flood frequency is well reproduced by the sediment record when considering both GB-I and GB-II. Indeed, both the three time periods and the range of flood-frequency values (from 1 to 4 per 11 years) are very similar between records. If GB-II are removed from the sediment record, the reconstructed flood frequency shows a more pronounced decrease over the last decades (orange line in Fig. 9) than in the historical record. This may support a flood trigger (during a period of high grazing activity) for GB-II. Overall, the good agreement with the historical data, when considering both GB-I and GB-II, supports that Lago Inferiore sediments are a good recorder of the decadal variability of past floods.

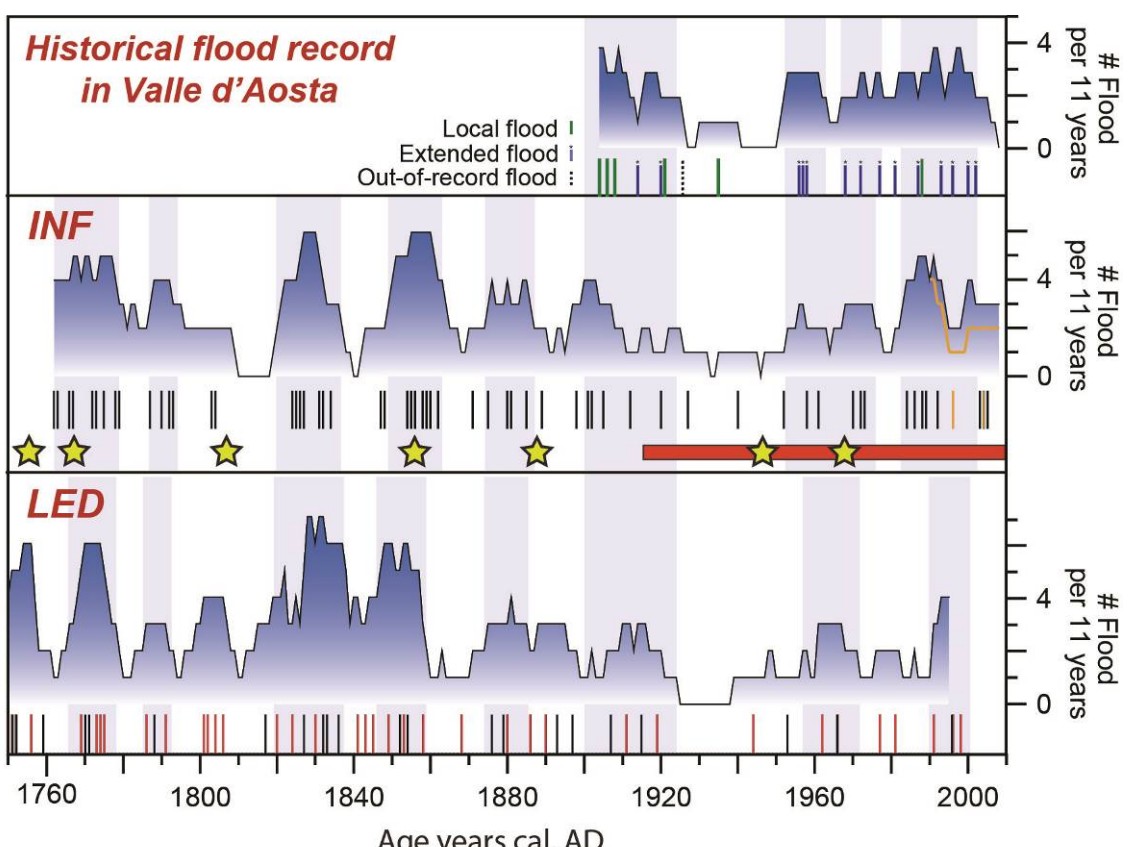

**Figure 9.** Comparison of the reconstructed Lago Inferiore de Laures (INF) flood frequency (11-years running sum) with the frequency (11-years running sum) of historical floods in Aosta Valley (Mercalli et al., 2003) and the frequency of summer-autumn floods recorded in Lago di Ledro (LED, Wirth et al., 2013). Vertical bars correspond to flood occurrences. For Lago Inferiore de Laures, the two orange vertical bars correspond to the GB-II. The orange curve corresponds to the flood frequency when these two deposits are not considered. Yellow stars correspond to the earthquake-induced deposits indicated as chronological markers and the horizontal red rectangle highlights the period dated by the 210Pb method. For Lago di Ledro record, black vertical bars correspond to summer floods and red vertical bars to autumn floods.

### 5.2.3. Paleoflood record in the regional climatic setting

Historical data revealed that flood events mostly occurred in summer and autumn (20 of 21), i.e. during the ice-free season of the lake. Hence, the variability of floods that impacted communities in Valle d'Aosta is well represented by the flood activity recorded in the Lago Inferiore sediment sequence. Among these events, 5 occurred in summer and early autumn and affected a localized area (i.e. only one mountain stream, Mercalli et al., 2003). According to the season and their limited spatial extent, these events are most probably triggered by local convective events, i.e. thunderstorms. The 15 other events occurred equally in summer and autumn and affected many tributaries and/or the main Dora Baltea River. As these events affected large catchments (ca. 200-2000 km²), they are most probably related to mesoscale convective events typical of the Mediterranean climate (e.g. Buzzi and Foschini, 2000). Thereby, the flood activity recorded in Lago Inferiore sediments is mainly related to large scale hydro-meteorological events and may represent a 'regional' signal of the past summer-autumn flood variability. As these mesoscale events are formed by humid air masses from the Mediterranean that flow northward through the Po Plain until the Alps (e.g. Buzzi and Foschini, 2000), they may also trigger floods in many Alpine regions located north of the Po Plain.

To test the 'regional' character of the reconstructed flood signal, the Lago Inferiore de Laures flood record was compared to the Lago di Ledro flood record. Lago di Ledro is a low-elevation lake (660 m a.s.l.) located 280 km east from Lago Inferiore de Laures, in the eastern part of the Alpine region located north to the Po Plain (Fig. 1). Floods in Ledro catchment (111 km²) also occur mainly in summer and autumn due to mesoscale convective events (Wirth et al., 2013). The extrapolation of the sedimentation rate enables to extend the centennial Lago Inferiore de Laures flood record to the last 270 years (Fig. 7). From the comparison with the Lago di Ledro flood record (Fig. 9), we observe that the range of flood-frequency values is in agreement between the two records, i.e. between 0 and 6 floods per 11 years. Secondly, we observe strong similarities in the two flood records with periods of high flood frequency in AD 1760-1780, 1785-1795, 1820-1835, 1875-1885, 1955-1975 and after 1990 and periods of low flood frequency in AD 1780-1785, 1810-1820, 1860-1875 and 1925-1955. However, some discrepancies between the two records can be noticed around AD 1800, 1890-1920 and 1980-1990. They may be related to localized events, e.g. thunderstorms, which may have different spatial and temporal dynamics between sites as evidenced by the record of several local floods between AD 1900 and AD 1910 (Fig. 9). Overall, there is a good agreement in the main trends of the flood frequencies, suggesting that the two flood records dominantly represent the decadal variability of mesoscale convective events triggering floods in this part of the Mediterranean Alps.

### 6. Conclusion

The high-resolution sedimentological study of Lago Inferiore de Laures revealed 77 beds that correspond to 76 event layers over the last ca. 270 years. A detailed analysis suggested that 8 of 76 event layers are related to 8 mass-movement events, while 66 of 76 are most probably related to flood events. The trigger of 2 event layers (those labelled GB-II) still remains uncertain. The temporal assignment suggests a flood trigger during a period of high grazing activity. However, further work is still required to confirm this hypothesis, e.g. by studying proxy of grazing activity like coprophilous fungal ascospores (e.g. Davis and Schafer, 2006; Etienne et al., 2013).

The 8 mass movements were chronologically compared to the well documented historical seismicity. The
comparison revealed that mass movements in Lago Inferiore de Laures are most probably triggered by strong
(epicentral MSK intensity of VI-IX) and/or close (distance to the lake of 25-120 km) earthquakes. Compared to
other Alpine lakes, the high frequency of earthquake-induced mass movements (8 over ca. 270 years) suggested
a high sensitivity of Lago Inferiore de Laures sediments to earthquake shaking. Indeed, this lake appeared to be
regionally the most sensitive with an ESTI value of 1.9, that may be explained by its high sedimentation rate.
The frequency of flood-induced deposits was compared to the frequency of historical summer-autumn floods
that affected mountain streams and rivers in Valle d'Aosta. This showed that the (multi-) decadal frequency of
flood events that impacted local populations is well reproduced by the sedimentary record. The comparison with
the flood record of Lago di Ledro, located 280 km east, suggested that the main trends of the (multi-) decadal
flood variability are in good agreement between records, suggesting a 'regional' character of the two
reconstructed flood signals linked to the typical Mediterranean mesoscale precipitation events.
Hence, this study showed that Lago Inferiore de Laures sediments seem to be a remarkable record of earthquakes
and floods, both natural hazards harming populations of this Alpine region. This should encourage further study
to extend the Lago Inferiore de Laures record further back in time. Such a long-term record of natural hazards
would improve our knowledge on the natural hazard occurrence and, thereby, enabling a better risk assessment.

**Acknowledgments**
B. Wilhelm's post-doctoral fellowship (2013-2014) was supported by a grant from the AXA Research Fund. We
would also like to thank Pierre Sabatier for his help to interpret the [210]Pb data. The authors thank Marteen Van
Daele and the anonymous reviewer for their constructive and helpful comments.

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
