# Peer review of "A multi-centennial record of past floods and earthquakes in Valle d'Aosta, Mediterranean Italian Alps"

_Natural Hazards and Earth System Sciences, 2016_

## Referee Comment (RC1) · M. Van Daele (Referee) · 12 Jan 2017

I reviewed the manuscript by Wilhelm et al. with pleasure, as my own research focus is also on lacustrine turbidites triggered by natural hazards, such as (mainly) earthquakes and floods. This manuscript is well written, well-structured and documented with figures of good quality. Apart from one main comment on the reasoning in the part where some of the deposits are linked to earthquakes, most of my comments and corrections are very minor, and listed below.

My main comment concerns the structure and reasoning in sections 5.1.2. and 5.1.3., and Figs 7 (and 8). Currently the GB-III, HB and MSB deposits are linked to "strong" and/or "close" earthquakes (lines 282-283) in section 5.1.2.. This seems to be rather

[Figure]

subjective. In section 5.1.3. the 'distance vs epicentral intensity' diagram (Fig 8A) is introduced, after which the authors conclude that the (subjectively) chosen earthquakes all plot above a certain threshold line. I believe the correct/objective way to do this, would be turning this reasoning around:

1) Firstly estimate for each historical earthquake MSK intensity at the lake (this could for example be the intercept of a line 'with the MSK intensity at 10 km from the lake' that runs through the earthquake and is parallel to the blue line (see also comment with Fig 7), or some other parameter that is linked to both epicentral intensity and distance.

2) Then plot those estimations on the time axis of Fig 7 (see also comments on Fig 7) and use that data to link deposits to a certain earthquake by projecting deposit ages on the time axis.

In principle I think that in order to do this, sections 5.1.2. and 5.1.3. should be swapped and therefore (partly) rewritten.

Minor comments:

Line 41: delete "been"

Line 50: THE magnitude (twice)

Line 53: "Reconstruction of past earthquake magnitudes AND EXTENT is .."? (or AND LOCATION or RUPTURE AREA)

Line 55: I suggest to add the example from New Zealand by Howarth et al. (2014), as this is an excellent study and example.

Line 63: "great earthquakes" are defined as M8-8.9, so I would avoid using "great" to describe an earthquake of unknown magnitude. Other defined descriptive words are: giant: M>9; major: M7-7.9; strong: M6-6.9; moderate: M5-5.9.

Lines 64-65: Hence, use "strong earthquakes" in Line 63?

Line 80: BEDrock? (twice)

Line 85: "...all the upper part of the catchment BARELY contributeS to the detrital...". I find "not" rather strong, as one cannot exclude that some of the very fine faction will not be trapped.

Line 87: delete "by"

Line 120: was this bath at room temperature or higher temp? How much was it diluted? Line 140: delete "the" before titanium

Line 148: remaining track changes

Lines 153-155: the 77 beds are a bit confusing, as there are actually only 76 horizons. The deformed layer is coeval with GB-IIIb, so these 2 beds correspond to only 1 event. This is a bit confusing, and should perhaps be clarified? It's also confusing in the abstract and the conclusions.

Line 173: GB-II beds seem to be intermediate between GB-I and HB. That's maybe worth mentioning in their description?

Line 187: "with MUCH LESS VARIATION OF THE median (D50)." There is definitely a noticeable variation in the D50

Lines 191-192: This correlation is not clear from Fig 3. The thick layer in INF13P4 is correlated to a layer below GB-IIIb in INF13P3.

Line 196: Too bad that 241 Am was not measured, as in the other papers the nuclear weapon tests are best represented by a peak in this isotope.

Line 206: refer to the original papers where the data was presented. Hence delete references to Wilhelm et al (2015, 2016), and add Wilhelm et al (2012) and Etienne et al (2013)

Line 215: "these distinct steps well mirror historical..."

Lines 220-221 and 227-228: repetition of nearly the same sentence

Lines 262-263: Keep as one paragraph. A new paragraph should not be started here

Line 266: "...as the result of strong earthquake shaking"

Line 283: I assume "1755" should be changed by "1767

Line 282-283: While there are only 9 estimated years between the deposits from 1780 and 1771, there are as much as 18 years between the correlated earthquakes from 1785 and 1767. Hence, the sedimentation rate in this interval should be half of that in the rest of the core. Is this plausible? If not, could it be that event GB-IIIe is erosional? As this the thickest graded bed in the record.

Lines 287-288: This statement should be supported by references: Monecke et al (2004) needs intensities of VI-VII for in situ deformation, while Moernaut et al (2004) already has lacustrine turbidites from intensities of V3/4 on (at least when they originate from a deltaic slope, which might be similar here), for turbidites from hemipelagic slopes intensities of VI1/2 are required. Van Daele et al (2015) finds turbidites (also from hemipelagic slopes) from intensities of VI on, while in-situ deformation is only found at an intensity of VII1/2. So these papers do indeed support this statement.

Lines 300-301: see main comment, but it would be good to actually estimate these ground motions in the lake area in some way.

Line 315: According to Fig 8 the ESTI is about 0.19 instead of 1.9

Lines 332-333: and similar to HB!

Line 343: Could you add a reference supporting this hypothesis?

Line 350: Mercalli et al (2003)

Lines 350-351: which year did this event occur? Could it anyway be indicated on Fig 9?

Line 357: "...frequently shows a more pronounced decrease over the..."

Line 360: "...sediments are a good recorder of flood variability."

Lines 373-374: "Hence, the variability of floods that impacted communities in Valle d'Aosta is well represented by the flood activity recorded in the Lago Inferiore sediment sequence."

Line 375- "...affected A localized area..."

Lines 374-377: Could these different types of flood events be indicated in Fig 9? This is important as it could explain the recent discrepancy (1980-1990) between the Lago Inferiore and Lago Ledro record, as the authors state in Line 395 that these discrepancies may be related to localized events such as thunderstorms (just as the 5 events from line 374). If the 1980-1990 discrepancy can indeed be explained by such events, than this will support the statement of the authors in Line 395.

Lien 383: "...north OF the Po Plain."

Line 392: "...periods of HIGH flood frequency..."

Lines 401-402: again confusing with the 77 layers for 76 events

Line 402 and 407: call it "8 mass-movement EVENTS". Because 1 mass-movement events might include several synchronous mass movements (especially when they are triggered by earthquakes".

Lines 412 and 417: some journals do not want references in the conclusions, as this should be the conclusions of this study, not any other. I personally do not have a big problem with it, but on the other had I also do not think it is crucial here.

Figures Fig 1: "HistoricAL earthquakes"

Fig 4: Both Q50 and D50 used. Keep it at D50 for each scale?

Fig 7: The correlation of the event beds to the historical earthquakes in this figure

should be done in a more objective way, as currently it is hard for the reader to review the correlation. I propose the following: - Make a vertical projection from all GB-IIIs, HB and MSB from the age model onto the time axis. This way the reader can see the estimated age of each bed. - Add above the (horizontal) time axis a new axis with ALL significant earthquakes and their age. Instead of simply mentioning each earthquake (which is rather subjective), earthquakes could be represented by a bar of which the height/length is determined by "the estimated MSK intensity of this earthquake in Lake Inferiore" (this MSK intensity could be estimated for each of these earthquakes, by pulling a line that (i) is parallel to the blue line in Fig 8, and (ii) crosses the red dot that represents that earthquake. The intercept of this line with the intensity axis at 10 km from the lake could represent the estimated MSK intensity). By doing this, a few earthquakes (at least 2, i.e. the black dots that are on or above the blue line in Fig 8) that are currently not shown on this figure, will also show up, even though they do not correlate to any of the graded beds. Alternatively (and I would personally prefer this option) the authors could even add some more earthquakes that are just below the blue line in Fig 8. These would have a shorter bar, and thereby it becomes clear that only earthquakes with the longest bar are represented by graded beds.

Fig 8: The black dots on and above the blue line should also have a date (or should at least be presented on the time axis in Fig 7). I assume one of them is the 1905 earthquake that is indicated on Fig 1?

Fig 9: indicate the one May flood and the different types of floods (limited vs large spatial extent) on the historical record.

References

Etienne, D., Wilhelm, B., Sabatier, P., Reyss, J.-L., Arnaud, F., 2013. Influence of sample location and livestock numbers on Sporormiella concentrations and accumulation rates in surface sediments of Lake Allos, French Alps. Journal of Paleolimnology 49, 117-127. doi: 10.1007/s10933-012-9646-x

Howarth, J.D., Fitzsimons, S.J., Norris, R.J., Jacobsen, G.E., 2014. Lake sediments record high intensity shaking that provides insight into the location and rupture length of large earthquakes on the Alpine Fault, New Zealand. Earth and Planetary Science Letters 403, 340-351. doi: 10.1016/j.epsl.2014.07.008

Monecke, K., Anselmetti, F.S., Becker, A., Sturm, M., Giardini, D., 2004. The record of historic earthquakes in lake sediments of Central Switzerland. Tectonophysics 394, 21-40.

Moernaut, J., Van Daele, M., Heirman, K., Fontijn, K., Strasser, M., Pino, M., Urrutia, R., De Batist, M., 2014. Lacustrine turbidites as a tool for quantitative earthquake reconstruction: New evidence for a variable rupture mode in south central Chile. Journal of Geophysical Research: Solid Earth, 2013JB010738. doi: 10.1002/2013JB010738

Van Daele, M., Moernaut, J., Doom, L., Boes, E., Fontijn, K., Heirman, K., Vandoorne, W., Hebbeln, D., Pino, M., Urrutia, R., Brümmer, R., De Batist, M., 2015. A comparison of the sedimentary records of the 1960 and 2010 great Chilean earthquakes in 17 lakes: Implications for quantitative lacustrine palaeoseismology. Sedimentology 62, 1466-1496. doi: 10.1111/sed.12193

Wilhelm, B., Arnaud, F., Enters, D., Allignol, F., Legaz, A., Magand, O., Revillon, S., Giguet-Covex, C., Malet, E., 2012. Does global warming favour the occurrence of extreme floods in European Alps? First evidences from a NW Alps proglacial lake sediment record. Climatic Change 113, 563-581. doi: 10.1007/s10584-011-0376-2

Wilhelm, B., Nomade, J., Crouzet, C., Litty, C., Sabatier, P., Belle, S., Rolland, Y., Revel, M., Courboulex, F., Arnaud, F., Anselmetti, F.S., 2016. Quantified sensitivity of small lake sediments to record historic earthquakes: Implications for paleoseismology. Journal of Geophysical Research: Earth Surface, n/a-n/a. doi: 10.1002/2015jf003644

Wilhelm, B., Sabatier, P., Arnaud, F., 2015. Is a regional flood signal reproducible from lake sediments? Sedimentology, n/a-n/a. doi: 10.1111/sed.12180

---

## Author Comment (AC1) · 30 Jan 2017

We thank very much the reviewer for his constructive comments. A point-by-point reply to the reviewer's comments can be found below, as well as the marked-up manuscript version. Our response to the review comments is marked in yellow. In addition, we have indicated all changes in the annotated version of the revised manuscript in yellow.

Response to the main comment

My main comment concerns the structure and reasoning in sections 5.1.2. and 5.1.3., and Figs 7 (and 8). Currently the GB-III, HB and MSB deposits are linked to "strong" and/or close" earthquakes (lines 282-283) in section 5.1.2.. This seems to be rather

subjective. In section 5.1.3. the 'distance vs epicentral intensity' diagram (Fig 8A) is introduced, after which the authors conclude that the (subjectively) chosen earthquakes all plot above a certain threshold line. I believe the correct/objective way to do this, would be turning this reasoning around: 1) Firstly estimate for each historical earthquake MSK intensity at the lake (this could for example be the intercept of a line 'with the MSK intensity at 10 km from the lake' that runs through the earthquake and is parallel to the blue line (see also comment with Fig 7), or some other parameter that is linked to both epicentral intensity and distance. 2) Then plot those estimations on the time axis of Fig 7 (see also comments on Fig 7) and use that data to link deposits to a certain earthquake by projecting deposit ages on the time axis. In principle I think that in order to do this, sections 5.1.2. and 5.1.3. should be swapped and therefore (partly) rewritten.

As the proposed approach seems indeed a good way to make our choices more 'objective', we follow the reviewer's recommendations and modify the manuscript and the figure accordingly. See sections 5.1.2. and 5.1.3. and Fig. 7.

Response to the minor comments: Line 41: delete "been" This has been changed as proposed.

Line 50: THE magnitude (twice) This has been changed as proposed.

Line 53: "Reconstruction of past earthquake magnitudes AND EXTENT is .."? (or AND LOCATION or RUPTURE AREA) This has been changed as proposed.

Line 55: I suggest to add the example from New Zealand by Howarth et al. (2014), as this is an excellent study and example. The reference has been added as proposed.

Line 63: "great earthquakes" are defined as M8-8.9, so I would avoid using "great" to describe an earthquake of unknown magnitude. Other defined descriptive words are: giant: M>9; major: M7-7.9; strong: M6-6.9; moderate: M5-5.9. Lines 64-65: Hence, use "strong earthquakes" in Line 63? This has been changed as proposed.

Line 80: BEDrock? (twice) This has been changed as proposed.

Line 85: ": : :all the upper part of the catchment BARELY contributeS to the detrital: : :". I find "not" rather strong, as one cannot exclude that some of the very fine faction will not be trapped. This has been changed as proposed.

Line 87: delete "by" This has been changed as proposed.

Line 120: was this bath at room temperature or higher temp? How much was it diluted? Requested information have been added: "in a temperate bath of diluted (30%) hydrogen peroxide"

Line 140: delete "the" before titanium This has been changed as proposed.

Line 148: remaining track changes Lines 153-155: the 77 beds are a bit confusing, as there are actually only 76 horizons. The deformed layer is coeval with GB-IIIb, so these 2 beds correspond to only 1 event. This is a bit confusing, and should perhaps be clarified? It's also confusing in the abstract and the conclusions. We clarified this point by indicating 76 event layers in the abstract and conclusions. In the main text, we kept the description of 77 beds (as observed) and add a sentence in the discussion part (section 5.1.1.) to highlight that 2 beds actually correspond to 1 event layer (l. 266-268).

Line 173: GB-II beds seem to be intermediate between GB-I and HB. That's maybe worth mentioning in their description? This is now mentioned.

Line 187: "with MUCH LESS VARIATION OF THE median (D50)." There is definitely a noticeable variation in the D50 This has been changed as proposed.

Lines 191-192: This correlation is not clear from Fig 3. The thick layer in INF13P4 is correlated to a layer below GB-IIIb in INF13P3. This has been nuanced as the graded bed becomes actually much thicker only in core INF13P3.

Line 196: Too bad that 241 Am was not measured, as in the other papers the nuclear

weapon tests are best represented by a peak in this isotope. Line 206: refer to the original papers where the data was presented. Hence delete references to Wilhelm et al (2015, 2016), and add Wilhelm et al (2012) and Etienne et al (2013) This has been changed as proposed.

Line 215: "these distinct steps well mirror historical: : :" This has been changed as proposed.

Lines 220-221 and 227-228: repetition of nearly the same sentence The first sentence has been removed.

Lines 262-263: Keep as one paragraph. A new paragraph should not be started here This has been changed as proposed.

Line 266: ": : :as the result of strong earthquake shaking" This has been changed as proposed.

Line 283: I assume "1755" should be changed by "1767 This has been corrected.

Line 282-283: While there are only 9 estimated years between the deposits from 1780 and 1771, there are as much as 18 years between the correlated earthquakes from 1785 and 1767. Hence, the sedimentation rate in this interval should be half of that in the rest of the core. Is this plausible? If not, could it be that event GB-IIIe is erosional? As this the thickest graded bed in the record. Indeed, such an abrupt change in the sedimentation rate seems to be unlikely. An erosional base for GB-IIIe is possible, also because it's the coarsest deposit (higher current energy). However, the stratigraphic correlation between cores INF13P3 and INF13P4 does not reveal clear evidence of erosion.

Lines 287-288: This statement should be supported by references: Monecke et al (2004) needs intensities of VI-VII for in situ deformation, while Moernaut et al (2004) already has lacustrine turbidites from intensities of V3/4 on (at least when they originate from a deltaic slope, which might be similar here), for turbidites from hemipelagic slopes

intensities of VI1/2 are required. Van Daele et al (2015) finds turbidites (also from hemipelagic slopes) from intensities of VI on, while in-situ deformation is only found at an intensity of VII1/2. So these papers do indeed support this statement. As this sentence was not necessary, it has been removed.

Lines 300-301: see main comment, but it would be good to actually estimate these ground motions in the lake area in some way. Line 315: According to Fig 8 the ESTI is about 0.19 instead of 1.9 This has been corrected.

Lines 332-333: and similar to HB! This has been changed as proposed.

Line 343: Could you add a reference supporting this hypothesis?

Line 350: Mercalli et al (2003) This has been changed.

Lines 350-351: which year did this event occur? Could it anyway be indicated on Fig 9? The year (AD 1926) was added in the manuscript and this event was indicated in Fig. 9. Line 357: ": : :frequently shows a more pronounced decrease over the: : :" This has been changed as proposed.

Line 360: ": : :sediments are a good recorder of flood variability." This has been changed as proposed.

Lines 373-374: "Hence, the variability of floods that impacted communities in Valle d'Aosta is well represented by the flood activity recorded in the Lago Inferiore sediment sequence." This has been changed as proposed.

Line 375- ": : :affected A localized area: : :" This has been changed as proposed.

Lines 374-377: Could these different types of flood events be indicated in Fig 9? This is important as it could explain the recent discrepancy (1980-1990) between the Lago Inferiore and Lago Ledro record, as the authors state in Line 395 that these discrepancies may be related to localized events such as thunderstorms (just as the 5 events from line 374). If the 1980-1990 discrepancy can indeed be explained by such events,

than this will support the statement of the authors in Line 395. The different types of floods have been indicated in Fig. 9. However, this does not allow explaining the 1980-1990 discrepancy because localized events may be recorded in Lago Inferiore but not in the historical data (e.g. if they do not affect the populations). In this sense, they are much more (30) events recorded over the last century in Lago Inferiore than (20) in the historical data.

Lien 383: ": : :north OF the Po Plain." This has been changed.

Line 392: ": : :periods of HIGH flood frequency: : :" This has been changed.

Lines 401-402: again confusing with the 77 layers for 76 events This has been changed (see previous comment on this point).

Line 402 and 407: call it "8 mass-movement EVENTS". Because 1 mass-movement events might include several synchronous mass movements (especially when they are triggered by earthquakes". This has been changed as proposed.

Lines 412 and 417: some journals do not want references in the conclusions, as this should be the conclusions of this study, not any other. I personally do not have a big problem with it, but on the other had I also do not think it is crucial here. These references have been removed.

Figures Fig 1: "HistoricAL earthquakes" This has been changed.

Fig 4: Both Q50 and D50 used. Keep it at D50 for each scale? This has been changed.

Fig 7: The correlation of the event beds to the historical earthquakes in this figure should be done in a more objective way, as currently it is hard for the reader to review the correlation. I propose the following: - Make a vertical projection from all GB-IIIs, HB and MSB from the age model onto the time axis. This way the reader can see the estimated age of each bed. - Add above the (horizontal) time axis a new axis with ALL significant earthquakes and their age. Instead of simply mentioning each earthquake (which is rather subjective), earthquakes could be represented by a bar of which the

height/length is determined by "the estimated MSK intensity of this earthquake in Lake Inferiore" (this MSK intensity could be estimated for each of these earthquakes, by pulling a line that (i) is parallel to the blue line in Fig 8, and (ii) crosses the red dot that represents that earthquake. The intercept of this line with the intensity axis at 10 km from the lake could represent the estimated MSK intensity). By doing this, a few earthquakes (at least 2, i.e. the black dots that are on or above the blue line in Fig 8) that are currently not shown on this figure, will also show up, even though they do not correlate to any of the graded beds. Alternatively (and I would personally prefer this option) the authors could even add some more earthquakes that are just below the blue line in Fig 8. These would have a shorter bar, and thereby it becomes clear that only earthquakes with the longest bar are represented by graded beds. The figure was modified as proposed by the reviewer, so that a chronicle of earthquakes expected to have triggered the largest ground motions in the lake area was added to make easier the comparison between ages of mass-movement deposits and dates of the 'strongest' historical earthquakes. See also answer to the main comment of the reviewer.

Fig 8: The black dots on and above the blue line should also have a date (or should at least be presented on the time axis in Fig 7). I assume one of them is the 1905 earthquake that is indicated on Fig 1? The date of the earthquake corresponding to the black dot above the blue line has been added because this event is now discussed in the manuscript.

Fig 9: indicate the one May flood and the different types of floods (limited vs large spatial extent) on the historical record. Everything has been added as proposed.

Please also note the supplement to this comment:
http://www.nat-hazards-earth-syst-sci-discuss.net/nhess-2016-364/nhess-2016-364-AC1-supplement.zip

2016.

---

## Referee Comment (RC2) · M. Van Daele (Referee) · 31 Jan 2017

I am very pleased to see that the authors followed practically all of my suggestions. Especially the addition of the local seismic intensity of the historical earthquakes (and their addition to Fig. 7) makes their correlation with the MSB, HB and GB-III deposits more robust.

I just have one additional suggestion. By differentiating between local and extended floods in the historical record (Fig. 9), it seems clear that the AD 1900-1910 discrepancy between Lake Inferiore and Lake Ledro is a result of local floods that influence Lake Inferiore, but not Lake Ledro. This correlation strongly supports one of the final statements in the discussion (lines 400-401): that these discrepancies may be linked

to such local thunderstorms. Hence, I propose to add something lake: "...as evidenced by the record of several local floods between AD 1900 and AD 1910 (Fig. 9)".

---

## Referee Comment (RC3) · Anonymous Referee #2 · 3 Mar 2017

**March 2017**

**A review of "A multi-centennial record of past floods and earthquakes in Valle d'Aosta, Mediterranean Italian Alps" by Bruno Wilhelm et al.**

**Manuscript number: nhess-2016-364**

This paper presents a lake sediment-based reconstruction of historical earthquakes and flood events in the Italian Alps over the last ~270 years. It builds on the burgeoning literature investigating palaeofloods and palaeoseismicity and its research objectives fit within the scope of NHESS. It is well-written, the data are generally analysed rigorously, the figures are largely clear and effective and I enjoyed reading the paper.

I have a number of interpretational queries and requests for clarification on certain aspects of the paper prior to recommending it for publication. In addition, I have a broader concern around the scope and impact of the paper. The authors and their collaborators have published a series of papers on this theme from various lakes in the European Alps over the last few years. If the authors are pitching it as a further case study, that's OK, and it meets the criteria of NHESS by presenting new data. On the other hand, the authors state on line 316 that Lago Inferiore has the "highest Earthquake Sensitivity Threshold Index of any studied Alpine lake". This is a much stronger statement than is made in the abstract. I urge the authors to consider re-framing the paper so they sell its novel aspects

**Important interpretational queries**

**(i)** The sediment accumulation rate is surprisingly high if the majority of the catchment is inactive, owing to the high-elevation lakes, and it is frozen for half the year. This will leave an active catchment in the order of 1-2 km$^2$. I suggest you elaborate further on the sources of sediment, especially how much may be glacially-derived material. Will this not have a very different sedimentological signal to the floods and mass movements?

**(ii)** It is unclear how you associate the lamination thickness with the grain size measurements sampled continuously at 5-mm intervals. In the Passega-type diagrams, what did you do if a maximum D50 or D80 value was derived from a 5-mm slice that overlapped into another distinguishable lamination?

**(iii)** The suggestion that the $^{137}$Cs spike associated with AD 1963 weapons testing has been diluted by the Chernobyl signal seems unlikely, considering the rate of sediment accumulation. AD 1963 should occur within a band at 12-18 cm (1σ), which corresponds with GB-IIIa. Is it more likely that this mass movement may have redeposited older material and diluted the atmospheric $^{137}$Cs signal?

**(iv)** The extrapolation of the age-depth model is a concern, although I appreciate this cannot be easily resolved without substantial effort e.g. acquiring radiocarbon ages. I presume the authors looked for earlier metal signals reflecting earlier industrial emissions and/or mining/smelting? Further, sediment density is higher below 20-cm. Could this point to greater input of clastic material? I suggest the authors make a convincing case that sediment accumulation rates are likely to have remained constant through this time window. In particular, would the SAR have remained constant as the glacier(s) in the catchment retreated after the mid-19$^{th}$ century maximum (assuming it followed the regional pattern)?

**(v)** The sedimentological evidence of mass movements is convincing but can anything be inferred from the different depositional characteristics of the four mass movement layers? Is the likelihood

for one depositional mechanism to occur sensitive to earthquake intensity or distance from epicentre, for example? Or does the lake and/or catchment evolution influence which type of mass movement deposit occurs in response to an earthquake? I've seen little on this in the literature and it would be an interesting point to try and make.

**(vi)** The role of glacial input and/or snow avalanches has not been considered fully. The former could make a significant contribution to the basal sediments because the active catchment from the eastern stream is so small. There is potential for snow avalanches to deliver a characteristic deposit – see some of the work by Eivind Støren and colleagues. This could a factor in the discussion on lines 332-334. Are there any records of avalanches in those years or local meteorological data that suggest particularly warm springs, which could have triggered widespread snowmelt? This notion of snowmelt applies more broadly, as the lake is frozen for 6 months of the year. Do the historical data (as referred to on line 372, presumably derived from Mercalli et al. 2003) suggest any regional floods triggered by snowmelt?

**(vii)** On Figure 8 there appears to be two earthquakes that plot above the sensitivity threshold. In terms of fully understanding the process sedimentology, I suggest the authors offer some explanation as to why those earthquakes did not leave a preservable imprint.

**(viii)** I am unconvinced by the argument that grazing facilitated thicker recent event deposits. Did grazing in the catchment really only begin in the 1990s? It would be helpful for the authors to provide evidence.

**Figure 1:** (i) The colour scheme associated with the DEM ought to be incorporated; (ii) The purpose of panel C is unclear. The lake appears disconnected from the major regional tributaries

**Figure 3:** (i) Could the horizontal layer stripes be shaded to reflect the different processes? (ii) Explain in the caption what the layer codes represent, or at least point the reader to the relevant section; (iii) The matrix-supported layer is very difficult to distinguish. Could you use a different colour scheme or patterning?

**Figure 4:** (i) Change 'sedimentary' to 'sediment' on the *y*-axis

**Figure 7:** Spell out what "$I_o$" and "$d$" are in the caption

**Figure 9:** (i) Spell out "INF" and "LED" or else include these codes in the caption; (ii) What does the horizontal red line represent?

**Minor comments**

Lines 34-35: the phrase on 'robust risk assessments' is rather vague

Line 36: Should include a reference

Line 38: "have been" instead of "were"

Line 46: "In **the** case of earthquakes…"

Line 49: "centur**ies**"

Line 61: remove "it"

Line 120: replace "during" with "from"

Line 140: remove "the"

Line 150: "…deposits**,** representing…"

Line 150: come up with a better technical word than "interrupted"

Line 197: add "down-core" or similar at the end of the sentence

Line 207: "cannot **be** as clear defined."

Global change: the word "decennial" is odd. I suggest a global change to "decadal"

Line 379 and section 5.2.3: I suggest the authors insert additional references to Mediterranean climate in this section (some of which are listed in the bibliography and referenced elsewhere)

---

## Author Comment (AC3) · 21 Mar 2017

We thank the reviewer for his comments. A point-by-point reply to these comments can be found below, as well as the marked-up manuscript version. Our response to the review comments is marked in yellow. In addition, we have indicated all changes in the annotated version of the revised manuscript in yellow.

Response to the main comment:

I have a broader concern around the scope and impact of the paper. The authors and their collaborators have published a series of papers on this theme from various lakes in the European Alps over the last few years. If the authors are pitching it as a

further case study, that's OK, and it meets the criteria of NHESS by presenting new data. On the other hand, the authors state on line 316 that Lago Inferiore has the "highest Earthquake Sensitivity Threshold Index of any studied Alpine lake". This is a much stronger statement than is made in the abstract. I urge the authors to consider re-framing the paper so they sell its novel aspects.

This aspect is already clearly stated in the abstract (l.20-22). We just avoid technical term (ESTI) in the abstract to make the statement clearer to a broad audience: "Compared to other lake-sediment sequences, Lago Inferiore de Laures sediments appear to be regionally the most sensitive to earthquake shaking, offering a great potential to reconstruct the past regional seismicity further back in time."

Important interpretational queries (i) The sediment accumulation rate is surprisingly high if the majority of the catchment is inactive, owing to the high-elevation lakes, and it is frozen for half the year. This will leave an active catchment in the order of 1-2 km2. I suggest you elaborate further on the sources of sediment, especially how much may be glacially-derived material. Will this not have a very different sedimentological signal to the floods and mass movements?

Sedimentation rate in INF is as high as for many other high-elevation lakes of the Alps (see Fig. 8B). In addition, we explain in the section 2 (l. 84-86) that the glacial material is mostly trapped by the two lakes located upstream. Only a rock-glacier is present in the 'real' catchment of the lake but there is no fine sediments that might be transported by the temporary stream and deposited in the lake. Finally, deposits triggered by floods or mass movements are all mainly made of detrital materials coming from the catchment that is uniformly made of eclogitic micaschist. That's why there is here no clear sedimentological or geochemical differences between them.

(ii) It is unclear how you associate the lamination thickness with the grain size measurements sampled continuously at 5-mm intervals. In the Passega-type diagrams, what did you do if a maximum D50 or D80 value was derived from a 5-mm slice that

overlapped into another distinguishable lamination?

For the great majority of layers, there is no problem of overlap (i.e. we generally got 1 measure of grain size per layer). For the very few thin layers that may be sampled together, the same grain-size parameters are assigned as we are not able to reduce the sampling step below 5mm.

(iii) The suggestion that the 137Cs spike associated with AD 1963 weapons testing has been diluted by the Chernobyl signal seems unlikely, considering the rate of sediment accumulation. AD 1963 should occur within a band at 12-18 cm ($1\sigma$), which corresponds with GB-IIIa. Is it more likely that this mass movement may have redeposited older material and diluted the atmospheric 137Cs signal?

As the hypothesis of the dilution effect is indeed uncertain and not necessary, we have removed this mention. The hypothesis of the reviewer on the role of the mass movement seems also unlikely because the Cs signal below the mass-movement deposit is null, whilst in this case we should expect low values (not null) corresponding to the beginning of the atmospheric nuclear weapon test in the 1950's.

(iv) The extrapolation of the age-depth model is a concern, although I appreciate this cannot be easily resolved without substantial effort e.g. acquiring radiocarbon ages. I presume the authors looked for earlier metal signals reflecting earlier industrial emissions and/or mining/smelting? Further, sediment density is higher below 20-cm. Could this point to greater input of clastic material? I suggest the authors make a convincing case that sediment accumulation rates are likely to have remained constant through this time window. In particular, would the SAR have remained constant as the glacier(s) in the catchment retreated after the mid-19th century maximum (assuming it followed the regional pattern)?

Because of the uncertainties of radiocarbon ages (around a century or even more), this would not improve this chronology. In addition, there is no older Pb (or metal) contamination. The increase of density is mainly related to the compaction effect that

triggers a lower porosity and then a lower water content (classic in recent sediments). As indicated above, the glacier fluctuations are not expected to significantly influence the sedimentation rate (SR) because of the two lakes upstream that act as sediment traps.

(v) The sedimentological evidence of mass movements is convincing but can anything be inferred from the different depositional characteristics of the four mass movement layers? Is the likelihood for one depositional mechanism to occur sensitive to earthquake intensity or distance from epicentre, for example? Or does the lake and/or catchment evolution influence which type of mass movement deposit occurs in response to an earthquake? I've seen little on this in the literature and it would be an interesting point to try and make.

Indeed, this issue is an exciting research perspective in palaeoseismology. We have previously explored this approach based on a comprehensive review of earthquake-induced mass movements in similar high-elevation alpine lakes (Wilhelm et al., 2016, JGR). However, this appeared unsuccessful for those lakes.

(vi) The role of glacial input and/or snow avalanches has not been considered fully. The former could make a significant contribution to the basal sediments because the active catchment from the eastern stream is so small. There is potential for snow avalanches to deliver a characteristic deposit – see some of the work by Eivind Støren and colleagues. This could a factor in the discussion on lines 332-334. Are there any records of avalanches in those years or local meteorological data that suggest particularly warm springs, which could have triggered widespread snowmelt? This notion of snowmelt applies more broadly, as the lake is frozen for 6 months of the year. Do the historical data (as referred to on line 372, presumably derived from Mercalli et al. 2003) suggest any regional floods triggered by snowmelt?

About the glacial inputs, see comments above. About the avalanche and the work of C. Vasskog et al. (colleague of E. Støren), this is an issue we know well. Avalanches

generally deliver unsorted debris (e.g. coarse sand and gravels mixed with fine sediments) that form particular deposits that we do not observe here. In some cases, this may exceptionally trigger mass movements. However, this requires an avalanche strong enough to break the ice cover and then disturb the slope sediments (e.g. Wilhelm et al. 2013). From the review of mass movements recorded in many alpine lakes, we observed that the earthquake is a much more probable trigger than such a strong avalanche (Wilhelm et al., 2016, JGR). In addition, we do not have any information related to past avalanches in the catchment, or in the region, to support even more this aspect. Mercalli et al. (2003) do not describe the hydro-meteorological conditions that induced floods.

(vii) On Figure 8 there appears to be two earthquakes that plot above the sensitivity threshold. In terms of fully understanding the process sedimentology, I suggest the authors offer some explanation as to why those earthquakes did not leave a preservable imprint.

Actually there is only 1 earthquake that does not let a visible imprint (AD 1905, see also Fig. 7 where this event is highlighted with a question mark). We do not have any plausible explanation of this phenomenon. That's why we are not able to further discuss this aspect.

(viii) I am unconvinced by the argument that grazing facilitated thicker recent event deposits. Did grazing in the catchment really only begin in the 1990s? It would be helpful for the authors to provide evidence.

The relation between grazing and erosion, and the potential influence on the flood record, is not an argument but a fact as shown for instance by Giguet-Covex et al. (2011). You may also see Giguet-Covex et al., 2014, Nature Communications or Brisset et al., 2017, Geology. At this stage, we cannot further elaborate on this aspect in INF because of the lack of data. That's why we conclude that "further work is still required to confirm this hypothesis" (l. 411) and propose a research way to approach this issue:

[Figure]

"studying proxy of grazing activity like coprophilous fungal ascospores" (l. 412).

Figure 1: (i) The colour scheme associated with the DEM ought to be incorporated; (ii) The purpose of panel C is unclear. The lake appears disconnected from the major regional tributaries

The purpose of panel C is to highlight the streams affected by the historical floods documented by Mercalli et al. (2003) that we used in our comparison with the sedimentary record. We add a few words in the caption to highlight this aspect: "the hydrological network of Vallee d'Aosta that is regularly affected by floods as documented by Mercalli et al. (2003). (l. 73-74). We added the connection between the lake outlet and this hydrological network.

Figure 3: (i) Could the horizontal layer stripes be shaded to reflect the different processes? (ii) Explain in the caption what the layer codes represent, or at least point the reader to the relevant section; (iii) The matrix-supported layer is very difficult to distinguish. Could you use a different colour scheme or patterning?

We modified the horizontal grey bars as suggested and we added the meaning of this code in the caption as well as a reference to the sections in the text. The matrix-supported layer is already clearly highlighted with its label (MSB) in Fig. 3 and a zoom on this layer is also presented in Fig. 4.

Figure 4: (i) Change 'sedimentary' to 'sediment' on the y-axis The modification has been done.

Figure 7: Spell out what "Io" and "d" are in the caption This has been added in the caption.

Figure 9: (i) Spell out "INF" and "LED" or else include these codes in the caption; (ii) What does the horizontal red line represent?

INF and LED are now explained in the caption as well as the meaning of the red rectangle. (it highlights the period dated by the 210Pb method)

Minor comments Lines 34-35: the phrase on 'robust risk assessments' is rather vague We slightly modified the sentence to clarify it.

Line 36: Should include a reference We added the reference of the IPCC (2013)

Line 38: "have been" instead of "were" This has been modified.

Line 46: "In the case of earthquakes. . ." This has been modified.

Line 49: "centuries" This has been modified.

Line 61: remove "it" This has been modified.

Line 120: replace "during" with "from" We did not find this reviewer's proposition appropriate.

Line 140: remove "the" This has been modified.

Line 150: ". . .deposits, representing. . ." This has been modified.

Line 150: come up with a better technical word than "interrupted" We kept this word as it well highlights the sudden occurrence these deposits in term of sedimentological features and processes. In addition, this word is often used in the literature.

Line 197: add "down-core" or similar at the end of the sentence This has been added.

Line 207: "cannot be as clear defined." This has been modified.

Global change: the word "decennial" is odd. I suggest a global change to "decadal" This has been modified all along the manuscript.

Line 379 and section 5.2.3: I suggest the authors insert additional references to Mediterranean climate in this section (some of which are listed in the bibliography and referenced elsewhere) References have been added as suggested.

Please also note the supplement to this comment:
http://www.nat-hazards-earth-syst-sci-discuss.net/nhess-2016-364/nhess-2016-364-

[Figure]

AC3-supplement.zip